# Upcycling of Pharmaceutical Glass into Highly Porous Ceramics: From Foams to Membranes

**DOI:** 10.3390/ma15113784

**Published:** 2022-05-25

**Authors:** Akansha Mehta, Khaoula Karbouche, Jozef Kraxner, Hamada Elsayed, Dušan Galusek, Enrico Bernardo

**Affiliations:** 1FunGlass—Centre for Functional and Surface Functionalized Glass, Alexander Dubček University of Trenčín, 91150 Trenčín, Slovakia; jozef.kraxner@tnuni.sk (J.K.); dusan.galusek@tnuni.sk (D.G.); 2Department of Industrial Engineering, Università degli Studi di Padova, 35131 Padova, Italy; hamada.elsayed@unipd.it (H.E.); enrico.bernardo@unipd.it (E.B.); 3École Polytechnique de Constantine, Constantine 25000, Algeria; ibtissam48@yahoo.fr; 4Refractories, Ceramics and Building Materials Department, National Research Centre, Cairo 12622, Egypt; 5Joint Glass Centre of the IIC SAS, TnU AD and FChFT STU, 91150 Trenčín, Slovakia

**Keywords:** alkali activation, adsorption, glass foam, upcycling

## Abstract

The present COVID-19 emergency has dramatically increased the demand for pharmaceutical containers, especially vials. End-of-life containers, however, cannot be easily recycled in the manufacturing of new articles. This paper presents some strategies for upcycling of pharmaceutical glass into various porous ceramics. Suspensions of a fine glass powder (70 vol%) are used as a starting material. Highly uniform cellular structures may be easily prepared by vigorous mechanical stirring of partially gelified suspensions with added surfactant, followed by drying and firing at 550–650 °C. Stabilization of the cellular structures at temperatures as low as the glass transition temperature (T_g_) of the used glass is facilitated by thermal decomposition of the gel phase, instead of viscous flow sintering of glass. This finding enabled the preparation of glass membranes (∼78 vol% open porosity), by direct firing of hardened suspensions, avoiding any surfactant addition and mechanical stirring. The powders obtained by crushing of hardened suspensions, even in unfired state, may be used as a low-cost sorbent for dye removal.

## 1. Introduction

The outbreak of COVID-19 has increased the awareness about sustainability and recycling aspects in the pharmaceutical field, mainly concerning the disposal of glass waste. In particular, the outbreak has demanded the production of large amounts of glass packaging, which are, due to their specific chemical composition, difficult to recycle [1,2].

For decades, pharmaceutical industries have relied on vials made of borosilicate glass, appreciated for its high chemical stability [3] and indefinite, in principle, recyclability [4,5]. It has been estimated that, in general, 6 tons of recycled glass directly save 6 tons of resources and reduce the emission of CO_2_ by 1 ton [6]. However, this is hardly realized for pharmaceutical glasses, synthesized from high purity feedstock and shaped into preforms (e.g., tubes and rods, to be later transformed into vials, syringes, etc.), in highly specialized plants [7].

Recycling in a strict sense (‘closed-loop’ mode), e.g., remelting for the obtainment of the original glass articles, is actually far from being fully applied [7]. Important environmental advantages, including avoided dispersion in the environment and reduction of the use of natural raw materials, however, may originate from the reuse of discarded glass as raw material for new marketable products (‘open-loop’ mode). If the revenues compensate the costs of the transformation operations, a highly sustainable waste management model would be configured. The difference between economic value of the new products and manufacturing costs represents an undoubtedly challenging factor; only if high, open-loop recycling can be properly seen as ‘upcycling’ [8].

Glass foams, due to their much higher stability compared to polymer foams, may be seen as a model product of open-loop glass recycling, starting from finely powdered material, subjected to viscous flow sintering with concurrent gas evolution from specific additives (‘foaming agents’) [7,9]. Their application as thermally insulating materials plays a fundamental role in their overall sustainability, since manufacturing costs could be balanced by energy savings achieved during a long service life. However, the cost of foaming agents and the relatively high processing temperature (850–900 °C), compared to the minimum temperature for viscous flow sintering (~700 °C, for soda-lime glass), remain an open issue [7].

The present investigation is based on recent findings, using an alternative method, where the viscous flow sintering is applied to consolidate cellular bodies prepared previously at nearly room temperature, according to a ‘gel casting’ method. ‘Green’ foams are prepared by intensive mechanical stirring of alkali-activated aqueous suspensions of glass powders with added surfactants, undergoing progressive gelation. Firing of dried suspensions may be performed at low temperatures (e.g., ~700 °C, for soda-lime glass) [10]. Although highly porous cellular structures are well defined already before the firing, some ‘reshaping’ of pores may occur, as an effect of the decomposition of the compounds responsible for gelation, remaining as binding phases upon drying.

The alkali activation of soda-lime glass as well as bioactive glasses [10,11] promotes the formation of calcium silicate hydrated (C-S-H) compounds as binding phases at the glass surface. The (partial) alkaline dissolution of glass determines the release of Ca^2+^ ions into the solution. Almost CaO-free pharmaceutical glass does not allow for the development of C-S-H compounds, but gelation of alkali-activated suspensions is still possible and applicable to the manufacturing of glass foams [12]. Due to its boro-alumino-silicate chemical formulation, the hardening of glass suspensions has already been attributed to the development of hydrated alkali alumino-silicate zeolite-like gels, analogous to those formed by other alkali-activated materials (AAMs), such as geopolymers [13]. In contrast to previous investigations on glass foams fired at 700 °C, the present one is dedicated to the exploration of these gels also as sintering promoters, at even lower firing temperatures, as well as binders in ‘unfired’ products. The nature of the gels is also shown to be sensitive to the processing conditions.

The present investigation also aims at an extension of possible applications, to facilitate an extensive reuse of discarded glass. Unfired products are tested as sorbents for organic dyes, recognized as key polluting agents of wastewater. In statistics, about 30% of the dye used in the textile process is deactivated, thus causing the discharge of dyes in wastewater at amounts of tens of mg/L [14,15]. These concentrations must be considered significant since, according to the Ecological and Toxicological Association of Dyes and Organic Pigments Manufacturers (ETAD), levels of dyes higher than 0.1–1 mg/L cause visual pollution [16,17].

The removal of coloring agents from wastewater of textile districts through different treatment techniques, such as coagulation/flocculation, ultrafiltration, biological degradation, adsorption, oxidation, and advanced oxidation processes, has been the object of numerous papers and specific reviews [18,19]. Adsorption methods, where adsorption of dyes is controlled not only by adsorbent microstructure such as surface area and porosity but also by the type and areal density of surface functional groups are the most widely employed. Several recent studies of water purification methods have made use of glass and its properties, but using discharged glasses is still not well studied. Successful dye removal with the use of unfired products from discarded glass, in analogy with other AAMs [20,21], appears as a highly promising method, since it combines the added value with low manufacturing costs.

In this regard, the present work aims at illustrating the challenges and opportunities open for the upcycling of pharmaceutical glasses via the gel casting technique. In comparison with the previous experiments, after alkali activation, the temperature and the amount of surfactant will be reduced, and then eliminated. In previous experiments, foams from alkali-activated pharmaceutical glass were consolidated at 700 °C. Highly uniform cellular structures will be easily prepared by vigorous mechanical stirring of partially gelified suspensions with added surfactant, followed by drying and firing at 550–650 °C. For the sake of sustainability, the strategies for the reduction of surfactant will also be studied for application in the construction industry. Further, the removal of alkali carbonates after the test in boiling water will also be deeply studied in the article. Lastly, the ‘unfoamed’ and ‘unfired’ hardened samples will be utilized as adsorbents for the elimination of methylene blue dye. Further efforts will be dedicated by including TiO_2_ in glass, which increases the surface area of composite and acts as an efficient photocatalyst.

## 2. Experimental

### 2.1. Materials and Methods

Pharmaceutical boro-alumino-silicate glass (referred as ‘BSG’; chemical composition: SiO_2_ = 72 wt.%, Al_2_O_3_= 7 wt.%, B_2_O_3_ = 12 wt.%, Na_2_O = 6 wt.%, K_2_O= 2 wt.%, CaO = 1 wt.%, BaO < 0.1 wt.%) from crushed discarded pharmaceutical vials, provided by the company Stevanato Group (Piombino Dese, Padova, Italy), was used as the starting material.

Glass vials were first dry ball milled and sieved to obtain particles with a diameter < 75 μm. Fine powders were later mixed with an aqueous solution containing 2.5 M NaOH/KOH (50 mol%/50 mol%) reagent grade, Sigma-Aldrich, Gillingham, UK), with a solid loading of 68 wt.%. The glass powders were chemically treated in the solution for 4 h, under low-speed mechanical stirring (400 rpm). After alkali activation, the obtained suspensions of partially dissolved glass powders were cast in closed polystyrene cylindrical molds and cured at 75 °C for 2 h.

To prepare glass foams, Triton X-100 surfactant (2–4 wt.%; t-octylphenoxypolyethoxyethanol, Sigma-Aldrich, Gillingham, UK) was added to the mixtures after curing and the mixture was subjected to intensive mechanical stirring (at 2000 rpm). Foamed suspensions were left at 40 °C for 24 h to complete the curing, yielding ‘green’ foams. After demolding from plastic containers, the green foams were heat treated at a heating rate of 10 °C/min, at 550–650 °C, with a holding time of 1 h.

Selected mixtures without Triton X-100 addition were not foamed, but were immediately cured at 75 °C and 40 °C. Some samples were sintered at 550 °C (‘low fired’), whereas others remained unfired. One variant involved, besides glass powders, TiO_2_ nano-particles (21 nm, Evonik Degussa GmbH, Essen, Germany), in an amount corresponding to 10 wt.% of the total solid in suspension. They were tested in bulk form as well as in powder form after crushing and sieving < 75 μm.

### 2.2. Characterization

The crystalline phases present in green and foamed samples were identified by X-ray powder diffraction (XRD, Bruker D8 Advance, Karlsruhe, Germany—CuKα radiation, 0.15418 nm, 40 kV–40 mA, 2θ = 10–70°, step size 0.05°, 2 s counting time). The chemical bonding was studied by infrared spectroscopy (FTIR, FTIR model 2000, Perkin Elmer, Waltham, MA, USA). The spectra were recorded in the 4000–500 cm^−1^ range collecting an average of 64 scans with 2 cm^−1^ resolution.

Porous samples were cut into cubes of about 10 mm × 10 mm × 10 mm, later employed for density measurements and compressive tests. The geometrical density (ρ_geom_) of samples was computed from the weight-to-volume ratios on regular blocks, after careful determinations of weights and dimensions utilizing an analytical balance and a digital calliper. The apparent density (ρ_app_) and the true density (ρ_true_) were evaluated using a helium pycnometer (Micromeritics AccuPyc 1330, Norcross, GA, USA), operating on bulk or finely crushed samples, respectively. The measured density values were used to calculate the amount of total, open, and closed porosity. The compressive tests were performed at room temperature on at least nine samples from each group, using a universal testing machine (Quasar 25, Galdabini S.p.a., Cardano al Campo, Italy), operating at a cross-head speed of 1 mm/min.

The morphological characterizations were performed by scanning electron microscopy (FEI Quanta 200 ESEM, Eindhoven, The Netherlands). The specific surface area, the total pore volume, and the pore size of the samples were evaluated by N_2_ physisorption at −196 °C (ASAP 2010, Micromeritics, Norcross GA, USA). The samples were degassed at 150 °C and the specific surface area (SSA) was calculated in the relative pressure (p/p_0_) range between 0.05 and 0.30, by applying the Brunauer–Emmett–Teller (BET) multipoint method.

Selected materials, in powdered form, were tested as sorbents for the methylene blue dye. The powders (50 mg) were placed in beakers containing 10 mL of the dye solution (100 mg/L), under stirring, and left for 60 min in the dark. The suspensions were subsequently irradiated by UV light (Hg lamp, λ = 366 nm, power = 125 W, Helios Italquartz S.R.L., Milan, Italy). The kinetics and change of concentration of the dye were recorded after removing the solid fraction by filtration of the UV-irradiated suspension through a cellulose filter (0.22 μm) at pre-defined time intervals and the absorption maximum at 660 nm was measured using a UV–Vis spectrophotometer (Jasco V570).

## 3. Results

### 3.1. Revision of Gel Casting Technology: Reducing Both Surfactant and Firing Temperature

In previous experiments, foams from alkali-activated pharmaceutical glass had been consolidated at 700 °C [12]. Extensive viscous flow densification of glass powders was not expected, since the temperature of 700 °C had been identified as the threshold for pressure-less sintering (50 °C above the dilatometric softening point, 650 °C) [22] of the used pharmaceutical glass. However, the open-celled structure of ‘green’ foams (shown in Figure 1a,b), from the intensive mechanical stirring of alkali-activated glass suspensions, was not fully preserved. Although the total porosity remained high (~70 vol%), the foams also contained a significant fraction of closed porosity (~15 vol%) [12]. This was attributed to the ‘fluxing’ action of the gel phase enriched in alkali from glass/solution interaction, which was responsible for the hardening at a temperature close to the room temperature, undergoing thermal decomposition.

With the present investigation, for the sake of sustainability, further evidence of the fluxing action was explored. Figure 1c,d confirms that firing at only 650 °C still enabled some consolidation, with the significant advantage of nearly complete preservation of open porosity. As shown in Table 1, the closed porosity was almost negligible (<3 vol%). Considering the new foams were nearly completely open-celled, the crushing strength of approximately 4 MPa was quite high, compared with that (~8 MPa) of previous foams fired at 700 °C. According to the Gibson–Ashby model [23], the compressive strength of cellular solid, σ_c_, is a complex function of relative density (ρ_rel_ = 1 − P, where P is the total porosity):σ_c_ ≈ σ_bend_·f(Φ,ρ_rel_) = σ_bend_ [C·(Φ·ρ_rel_)^3/2^ + (1 − Φ)·ρ_rel_]
where σ_bend_ is the bending strength of the solid phase, C is a dimensionless calibration constant (~0.2), and the quantity (1 − Φ) expresses the contribution of the solid positioned in cell walls of closed pores. For an open-celled structure (Φ ∼ 1), the observed crushing strength of ~4 MPa could be correlated to a bending strength exceeding 100 MPa (see Table 1), well above the measured values for pore-free sintered pharmaceutical glass [22].

As shown by Table 1, a limited, but not negligible consolidation of ‘green’ foams occurred at even lower temperature, i.e., at 550 °C, corresponding to the transition temperature (T_g_) of pharmaceutical glass. Any possibility of viscous flow of the glass was therefore excluded, as a further proof of the binding action of the gel formed upon alkali activation. Figure 2a illustrates effective joining of adjacent glass particles in foamed bodies. The compressive strength of foams was degraded, but the obtained materials could still be considered for application in components, which do not have to withstand severe load, such as catalysts supports (as an example, alumina foams, for the same application, have comparable strength-to-density ratio [24]).

To further improve the sustainability, the surfactant addition was first reduced, and then completely eliminated. The reduction to 2 wt.% had a limited impact on the pore generation, since the green foams were only slightly less porous than those prepared with a 4 wt.% surfactant addition. Interestingly, there was a remarkable increase of strength already in the green state (0.7 MPa, corresponding to a bending strength of the solid phase of ~30 MPa). The presence of surfactant evidently affected the nature of the gel phase.

The last observation is supported by the results of infrared spectroscopy, as illustrated in Figure 3. The spectra of various variants of hardened glass suspensions after drying (the initial surfactant addition of 2–4 wt.%, no addition of surfactant) are shown. In all cases, hydration was confirmed by the bands at 3400 cm^−1^, O-H stretching, and 1600 cm^−1^, SiOH stretching [25]; the addition of surfactant is documented by the band at about 2800 cm^−1^ attributed to C-H_2_ stretch vibrations [10]. A marked difference of the main band, centered at 1000–1050 cm^−1^ and corresponding to tetrahedral stretching modes of Si-O (Si) and Si-O (Al) bridges was observed. The ‘main bonds’ in the network of the studied glass were significantly altered in foamed suspensions, the effect being more pronounced at increasing surfactant addition. The asymmetry of the peak at 1000–1050 cm^−1^ could be attributed to an increased amount of BO_4_ tetrahedral units (with B-O stretching vibration corresponding to a band at about 900 cm^−1^) compared to BO_3_ trigonal units (with B-O stretching vibration attributed to a band at about 1200 cm^−1^) [25,26]. No differences were noted after firing at 550 °C (we report a single spectrum, collected for a foam after firing); the residue of thermal decomposition of the gel (of any nature) was likely included in the glass, altering the structure (as demonstrated by the shape of the main band positioned at 1000–1050 cm^−1^).

An additional proof of the changes in the nature of the gel related to the surfactant addition was provided by X-ray powder diffraction analysis (Figure 4a). Previous investigations on foams [11] had indicated the formation of traces of zeolite phases (such as gmelinite, fajausite, and paragonite) after activation with NaOH. The activation with the NaOH/KOH mixture led to amorphous materials, featuring, according to NMR studies, mainly AlO_4_ and BO_4_ units. In the present case, foamed suspensions featured traces of hydrated sodium carbonate (Na_2_CO_3_·H_2_O, PDF#70-2148); this phase was accompanied by a newly formed amorphous gel, considering the shift (at 2θ = 27–37°) of the ‘amorphous halo’ compared to the as-received glass. Activated glass suspensions, dried without foaming, yielded both sodium and potassium-based hydrated carbonates, such as trona (Na_3_(CO_3_)(HCO_3_)·2H_2_O, PDF#29-1447) and kalicinite (K_2_CO_3_·H_2_O PDF#12-0292).

Hydrated carbonate phases were not actually responsible for the hardening of the activated glass suspensions. In fact, the observed gelation could be due to the overlapping contributions of hydrated carbonates embedded in an amorphous gel matrix at the surface of glass particles (Figure 5a,b). This was supported by the different behavior of samples, when put in boiling water for 15 min. ‘Green’ foams (foams from hardened suspensions, not stabilized by firing) completely disintegrated, whereas denser bodies (hardened suspensions not subjected to any surfactant addition and foaming) survived. Since hydrated alkali carbonates are known for their high solubility in water, the different behavior could be attributed to the gel. In foams, the incorporation of alkali ions probably did not simply stabilize AlO_4_ and BO_4_ units in the gel, but also promoted the formation of a highly depolymerized network, preventing the condensation of silanol groups and formation of strong Si-O bridges. In denser bodies, alkali ions likely entered the gel to a much lower extent, so that glass particles were effectively ‘glued’ together by condensation of silanol groups.

Figure 4b confirms the removal of alkali carbonates after the test in boiling water; the dissolution is also confirmed by a slight increase of the total porosity, as reported in Table 1. A strong bond between particles was also confirmed by the comparable compressive strength of the samples before and after the boiling test. The presence of a thin surface layer of a gel, responsible for the real hardening, is finally documented in Figure 5c,d. The glass particles were bound to each other (Figure 5c) and the surface of each particle was highly textured (Figure 5d)

An interpretation of the observed phenomena will probably be possible only by systematic studies of the gel structure, which will be the object of future studies. A possible preliminary explanation may come from the modification of the suspension/atmosphere interaction related to the presence of the surfactant, which affects two concurrent phenomena: the previously mentioned condensation of silanol groups and alkali incorporation. The surfactant, deposited at cell walls, may have prevented the removal of alkali ions from the gel by absorption of carbon dioxide from the atmosphere and carbonate formation.

‘Unfoamed’ samples, according to the abundant open porosity, appear as a highly promising material for membranes, e.g., for filtration of contaminated waters [27]. Firing at 550 °C is related to the densification facilitated by the liquid from the decomposition of surface compounds, rather than on viscous flow sintering of glass (see Figure 2b). However, such firing is in principle not necessary, since the simple hardening at a temperature close to the room temperature (‘cold consolidation’) yields samples nearly equivalent in terms of crushing strength and overall porosity, with the additional advantage of complete permeability (negligible closed porosity, as reported in Table 1).

### 3.2. Application of Cold Consolidated Pharmaceutical Glass

Alkali-activated materials have a recognized potential for the absorption of organic dyes [28,29]. Although the hardening of activated glass suspensions did not correspond to the formation of a homogeneous gel, it was interesting for the possible utilization of the material in water purification. A ‘green’ (unfoamed) body was crushed to fine powder and sieved below 75 μm. Such operation was intended to yield loose glass particles or small agglomerates of the glass powder, coated by a hydrated gel layer. The results in Table 2 indicate that the activation, yielding hydrated gel, increased the specific surface area compared to that of glass powders obtained by crushing.

Figure 6a shows the absorbance spectrum of a methylene blue solution (100 mg/L) in deionized water. The absorbance maximum at 660 nm was considered at the reference for the initial concentration (C_0_); the sensitivity to UV degradation of the specific organic species is inferred from the decrease in absorbance after 30 min irradiation (reduction of ~20% in the intensity of the absorbance peak; C/C_0_ = 0.78). Figure 6a also demonstrates that the reduction of the content of methylene blue could be enhanced through addition of granules of activated glass suspended in the solution. After 30 min irradiation and removal of powders by cellulose filter, the absorbance peak was reduced by 34% (C/C_0_ = 0.66). Longer exposition times, up to 80 min resulted in a nearly undetectable absorbance indication almost complete removal of the dye from the solution (Figure 6b).

The use of the activated BSG powder in methylene blue solution was accompanied by the dissolution of hydrated carbonates, as shown in Figure 4b. The dissolution of carbonates did not impair the sorption capability of the granulate, which was preserved through several sorption cycles (Figure 5c). The sorption efficiency was determined from the optical measurements of the solutions subjected to the process consisting of powder casting, exposition to UV light (30 min), and centrifugation, using the equation:$$\mathrm{Efficiency}=\frac{{\left(\frac{C}{C_{o}}\right)}_{\mathrm{cycle}\;n}}{{\left(\frac{C}{C_{o}}\right)}_{\mathrm{cycle}\;1}}\cdot 100\%$$
where $C_{o}$ is the initial concentration of the dye solution, C is the concentration of dye solution after irradiation time. The powder was subjected to 5 cycles to test the stability of its sorption capacity. The BSG activated powders retained nearly 90% of the initial performance after 5 cycles.

Future efforts will be dedicated to further extensions of the current research, i.e., the use of an activated pharmaceutical glass as a binder for a new generation of mortars, in analogy with current efforts with soda-lime glass [30], as an alternative construction material. Its binding action, however, can also be exploited for functional applications. Figure 7a shows the detail of a ‘composite’ granule, from the crushing of an activated suspension comprising pharmaceutical glass particles and TiO_2_ nanoparticles, revealing extensive gel formation between glass particles. This led to a significant increase of specific surface area, up to 95 m^2^/g, as reported in Table 2, but it did not compromise the stability (blocks of hardened suspensions did not undergo dissolution in boiling water). Figure 7b testifies that TiO_2_ was mostly present as the anatase polymorph (PDF#78-2486), with traces of rutile (PDF#76-0318); traces of hydrated sodium carbonate (Na_2_CO_3_·H_2_O) could also not be excluded. Finally, Figure 7c illustrates the synergy between sorbent matrix and TiO_2_ photocatalyst: composite powders (involving 10 wt.% TiO_2_) were more effective than pure TiO_2_ powders and, more importantly, led to an almost complete degradation of methylene blue after only 50 min of exposure.

The present investigation discloses new opportunities for a specific waste glass. The low processing temperatures, limited amounts of required additives, and a wide range of applications all meet the conditions for upcycling of the BSG glass.

## 4. Conclusions

Along with manufacturing of glass foams with the use of gel casting, alkali activation of pharmaceutical boro-alumino-silicate glass represents the opportunity for a much broader range of applications. The phases yielded by the interactions between glass and the alkaline solution play a multiform role. Upon heating, they yield a liquid phase that enables sintering of glass at much lower temperatures than the glass softening temperature required for viscous flow sintering. More importantly, they facilitate the consolidation of glass without firing; the separation of alkalis in the form of hydrated carbonate phases appears to be favorable for the stability of gels, which act as binders for glass particles.

The materials prepared by cold-consolidation of pharmaceutical glass powders can be used for the production of a new generation of glass membranes. They may also find applications as sorbents for photodegradation of dyes, also with the inclusion of a TiO_2_ photocatalyst. The range of products for high added value applications represents an economically viable way of valorization of discarded pharmaceutical glass.

## Figures and Tables

**Figure 1 materials-15-03784-f001:**
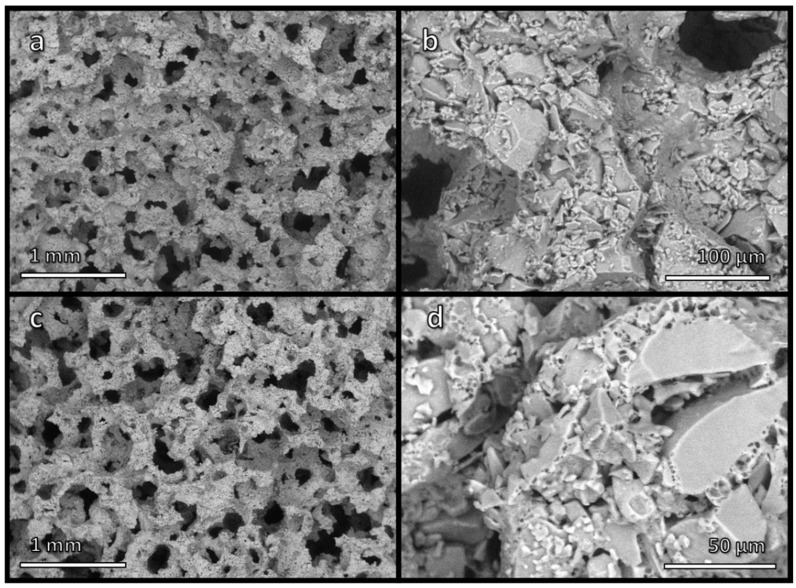
Low and high magnification micrographs of (**a**,**b**) foamed suspension of glass particles in alkaline solution, after drying; (**c**,**d**) glass foam, after firing at 650 °C.

**Figure 2 materials-15-03784-f002:**
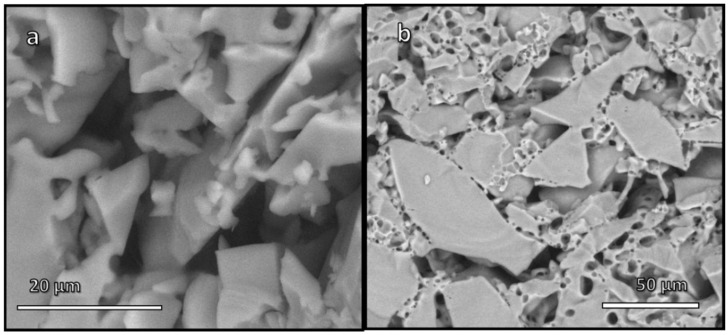
Microstructural details of porous materials from low temperature (550 °C) firing: (**a**) cell wall of a foam; (**b**) membrane.

**Figure 3 materials-15-03784-f003:**
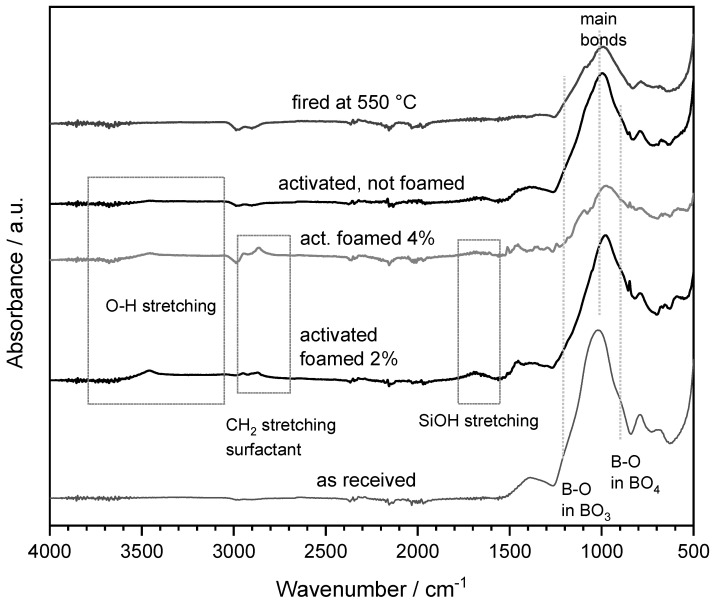
Fourier transform infrared (FTIR) analysis of boro−alumino-silicate glass in the as-received state, after activation and after firing.

**Figure 4 materials-15-03784-f004:**
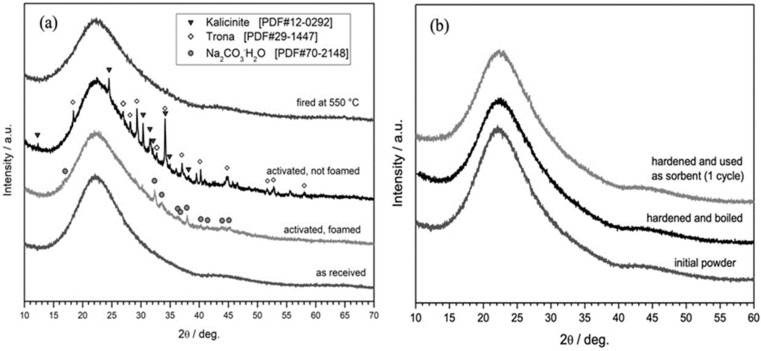
X-ray diffraction patterns of boro-alumino-silicate glass (**a**) in the as-received state, after activation and after firing; (**b**) after boiling and use as sorbent.

**Figure 5 materials-15-03784-f005:**
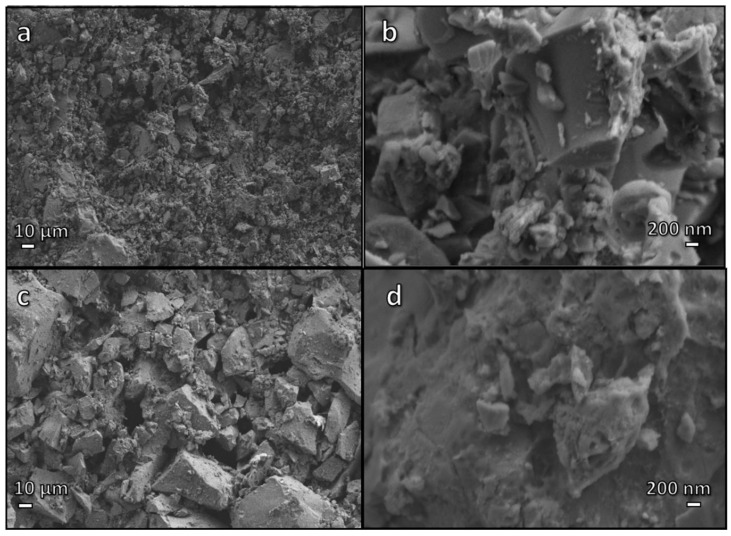
Low and high magnification micrographs of samples after cold consolidation: (**a**,**b**) simply hardened suspension; (**c**,**d**) after boiling test.

**Figure 6 materials-15-03784-f006:**
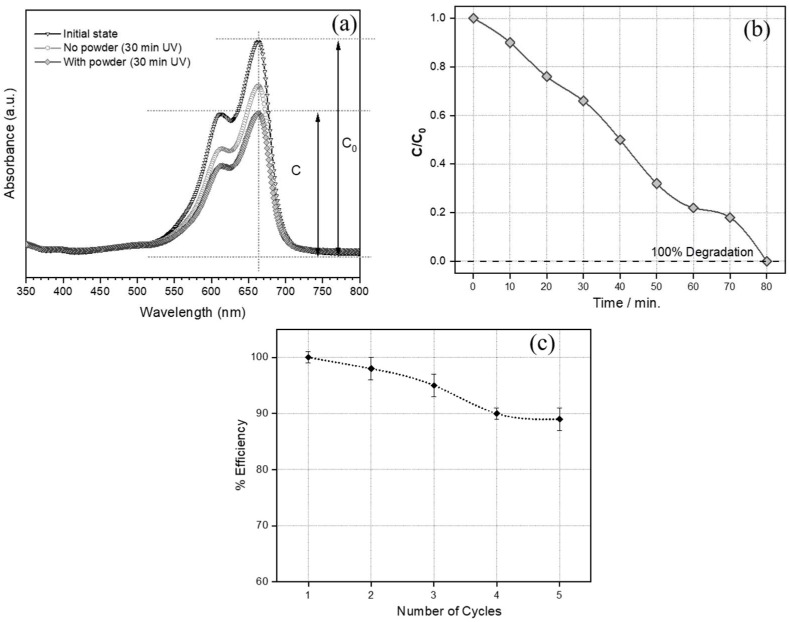
(**a**) Optical spectra of the methylene blue solution; (**b**) evolution of relative concentration of methylene blue with increasing UV exposition time; (**c**) efficiency of dye removal of the BSG activated powder with the number of sorption cycles.

**Figure 7 materials-15-03784-f007:**
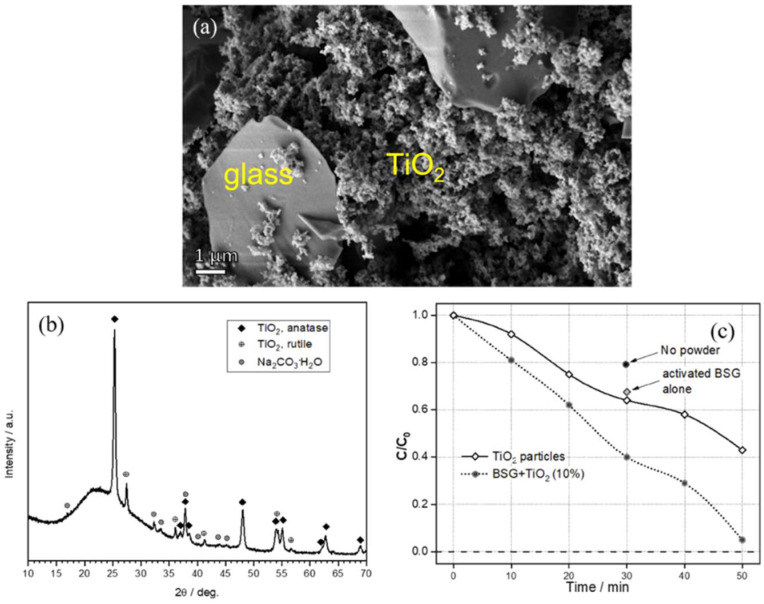
(**a**) Detail of glass/TiO_2_ composite granule; (**b**) diffraction analysis of hardened glass/TiO_2_ suspension; (**c**) evolution of relative concentration with increasing UV exposition time operating with TiO_2_ photocatalyst.

**Table 1 materials-15-03784-t001:** Physical and mechanical properties of porous materials from pharmaceutical glass.

Surfactant (wt.%)	4			2			(No Foaming)			
Firing T (°C)	650	550	green	650	550	green	650	550	green	green, after boiling
ρ_geom_ (g/cm^3^)	0.70 ± 0.03	0.54 ± 0.02	0.58 ± 0.03	0.77 ± 0.02	0.64 ± 0.03	0.57 ± 0.04	1.49 ± 0.05	1.41 ± 0.04	1.43 ± 0.10	1.32 ± 0.10
ρ_apparent_ (g/cm^3^)	2.17 ± 0.05	2.35 ± 0.05	2.31 ± 0.05	2.08 ± 0.05	2.36 ± 0.05	2.32 ± 0.05	2.01 ± 0.05	2.19 ± 0.05	2.33 ± 0.05	2.31 ± 0.05
ρ_true_ (g/cm^3^)	2.36 ± 0.05	2.37 ± 0.05	2.36 ± 0.05	2.37 ± 0.05	2.37 ± 0.05	2.36 ± 0.05	2.35 ± 0.05	2.36 ± 0.05	2.38 ± 0.05	2.37 ± 0.05
Total porosity (%)	70.3	75.4	77.2	67.5	72.9	75.8	36.5	40.2	38.7	42.5
Open porosity (%)	67.7	74.9	77.0	63.0	72.8	75.4	25.9	35.6	38.3	42.5
Closed porosity (%)	2.6	0.2	0.5	4.5	0.1	0.4	10.6	4.6	0.4	0
σ_comp_ (MPa)	3.9 ± 0.1	0.8 ± 0.1	0.5 ± 0.1	2.1 ± 0.1	0.7 ± 0.1	0.7 ± 0.1	19.4 ± 0.8	16.4 ± 0.8	21.3 ± 0.8	19.4 ± 0.8
σ_bend_ (MPa)	~120	~35	~20		~25	~30				

**Table 2 materials-15-03784-t002:** Results of BET characterization of BSG powders.

Sample	BET Surface Area (m^2^/g)	Pore Volume (cm^3^/g)	Pore Diameter (nm)
BSG, starting powders	3.4 ± 1	0.014	2.9
BSG activated, hardened, and crushed	16.0 ± 1	0.028	2.2
BSG activated, hardened, and crushed, after boiling	2.5 ± 1	0.015	2.9
BSG + TiO_2_ activated, hardened, and crushed	95.5 ± 1	0.146	2.2

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
