# Peer review of "Upcycling of Pharmaceutical Glass into Highly Porous Ceramics: From Foams to Membranes"

_materials, 2022, doi:10.3390/ma15113784_

Round 1

Reviewer 1 Report

This manuscript investigated the recycling of pharmaceutical glass by using alkali activation first. The alkali activation pretreatment helps the sintering at lower temperatures and facilitates the consolidation. The materials prepared by the glass powders could potentially be used for photodegradation of dyes with the inclusion of TiO2 content. The manuscript was recommended for publication after solving the following concerns.

(1) As for recycling, what are the common contaminations on the glass and how would them influence the recycling process?

(2) The title of the manuscript is confusing considering membranes are usually not porous. It should be revised.

(3) In Fig. 7a, glass and TiO2 should be marked on the image.

(4) In Table 1, at lower temperature of 550 C, why the sample has higher porosity but low closed porosity ratio?

Author Response

We thanks the editor and the reviewers for the comments on our manuscript:

Upcycling of Pharmaceutical Glass into Highly Porous Ceramics: From Foams to Membranes: materials-1725051

 by Akansha Mehta, Khaoula Karbouche, Jozef Kraxner, Hamada Elsayed, Dušan Galusek and Enrico Bernardo

We are pleased to submit a revised version. The changes are highlighted in red font. Our specific replies (R.=reviewer; A.=authors) are reported below.

 Reviewer #1

This manuscript investigated the recycling of pharmaceutical glass by using alkali activation first. The alkali activation pretreatment helps the sintering at lower temperatures and facilitates the consolidation. The materials prepared by the glass powders could potentially be used for photodegradation of dyes with the inclusion of TiO2 content. The manuscript was recommended for publication after solving the following concerns.

R1. Obs. #1 As for recycling, what are the common contaminations on the glass and how would them influence the recycling process?

A In the present work, ‘fresh’ glass vials were obtained from Stevanato Group. They corresponded to defective vials, not sent to pharmaceutical industries, and abundantly available, considering the huge overall production of vials. The idea behind this work is to set a direction of recycling the specific glass beyond remelting.  We are actually studying the effect of contaminants in these days. We are discussing with the company the reuse of components made with same glass, but coupled with metals and plastics; the internal crushing facility is currently separating metals and plastic, but the glass residues still has heterogeneities. Current cold consolidation tests do not show any particular effect of these heterogeneities, but they must be properly finalized.

R1. Obs. #2 The title of the manuscript is confusing considering membranes are usually not porous. It should be revised.

A We think that the title is appropriate, since we pass from macroporous glass foams to porous glass components based on the binding of glass particles with mesoporous gel. The porosity of the gel is finally exploited in the sorption experiments.

R1. Obs. #3 In Fig. 7a, glass and TiO2 should be marked on the image.

A We added the details in the specific figure.

R1. Obs. #4 Table 1, at lower temperature of 550 C, why the sample has higher porosity but low closed porosity ratio?

A The evolution of samples with firing is complicated. We have contasting effects, such as viscous flow of liquid phase determined by the decomposition of (alkali-enriched) gel, and gas evolution (from decomposition of hydrated and carbonated species).

Reviewer 2 Report

This paper presents some strategies for upcycling of pharmaceutical glass into various porous ceramics. From Foams to Membranes. The topic is novel. While there are some questions should be solved.  

  1. The introduction has written a lot, but it does not explain the direct relationship with this paper, what problem is this paper trying to solve, and what work has been carried out. A paragraph is required to link the background to the work of the thesis.
  2. For experimental, the chemical composition of Al2O3, B2O3, etc. is still wt%? it should be written clearly.
  3. In Figure 4a, the PDF standard card of three phases should be list inside the figure.
  4. It seems that Fig 5a and Fig 5c may come from one SEM picture. Beside, Fig 5b and Fig 5d may come from one SEM picture. The authors should check the experimental data carefully.

Author Response

we thanks the reviewers for the comments on our manuscript:

Upcycling of Pharmaceutical Glass into Highly Porous Ceramics: From Foams to Membranes: materials-1725051

 by Akansha Mehta, Khaoula Karbouche, Jozef Kraxner, Hamada Elsayed, Dušan Galusek and Enrico Bernardo

We are pleased to submit a revised version. The changes are highlighted in red font. Our specific replies (R.=reviewer; A.=authors) are reported below.

Reviewer #2

This paper presents some strategies for upcycling of pharmaceutical glass into various porous ceramics. From Foams to Membranes. The topic is novel. While there are some questions should be solved.  

R2. Obs. #1 The introduction has written a lot, but it does not explain the direct relationship with this paper, what problem is this paper trying to solve, and what work has been carried out. A paragraph is required to link the background to the work of the thesis.

A We are grateful to reviewer for the suggestion. We added the additional paragraph in the introduction section.

R2. Obs. #2 For experimental, the chemical composition of Al2O3, B2O3, etc. is still wt%? it should be written clearly.

A We added the details in the experimental section of the manuscript.

R2. Obs. #3 In Figure 4a, the PDF standard card of three phases should be list inside the figure.

A Figure 4a now contains the references to the PDF standard cards.

R2. Obs. #4 it seems that Fig 5a and Fig 5c may come from one SEM picture. Beside, Fig 5b and Fig 5d may come from one SEM picture. The authors should check the experimental data carefully.

A We agree with the reviewer concern, The micrographs in Fig. 5a and Fig. 5c belongs to same sample, but Fig. 5a micrograph is from simply hardened suspension and Fig. 5c micrographs is after boiling test of hardened suspension. The same is for Fig 5b and Fig 5d.

Reviewer 3 Report

In their study, “Upcycling of Pharmaceutical Glass into Highly Porous Ceramics: From Foams to Membranes”, the authors propose a method for recycling medical borosilicate glass (BSG) into adsorbing material capable of wastewater removal. Proposed procedure includes milling of BSG, further activation by strongly alkaline solution leading to partial silica gel-formation and, finally, preparation of membranes by adding surfactant to BSG powder with further annealing. Interestingly, the authors found that the annealing stage can be omitted and the membranes can be obtained even at nearly room temperature (cold consolidation). Finally, they apply these membranes and their composites with crystalline TiO2 for photodegradation of methylene blue.

This paper presents an accurate and skillful study based on an interesting idea of reusing medical glass waste. However, the following issues need to be clarified before the acceptance of the paper.

  1. Achilles' heel of this study is the BSG source choice. The authors supposed that the proposed procedure could be used to utilize medical glass waste. Actually, such glass waste is heavily polluted by organic biological materials and body liquids. Treating such pollutions with harsh alkaline solutions and annealing them can lead to unforeseen consequences, e.g. burning of organic phase, ash formation, glass cracking, foaming, etc. Such effects can not be considered when using new, fresh and shine medical glass. In turn, recycling of fresh medical-quality glass is nonsense.

To demonstrate viability of the proposed procedure the authors need to repeat their material preparation process using medical borosilicate glass polluted by some kind of model organic waste or, if possible, using actual waste medical glass.

  1. The authors report full methylene blue (MB) photodegradation by prepared BSG membrane even at the absence of TiO2 Is it MB’s self-photodegradation or membrane itself acts as a photocatalyst or all MB is just adsorbed by the membrane? Please explain.
  2. Part “3.1 From foams to membranes” lacks a description of its purpose.

Author Response

We thanks the reviewers for the comments on our manuscript:

Upcycling of Pharmaceutical Glass into Highly Porous Ceramics: From Foams to Membranes: materials-1725051

 by Akansha Mehta, Khaoula Karbouche, Jozef Kraxner, Hamada Elsayed, Dušan Galusek and Enrico Bernardo

We are pleased to submit a revised version. The changes are highlighted in red font. Our specific replies (R.=reviewer; A.=authors) are reported below.

Reviewer #3

In their study, “Upcycling of Pharmaceutical Glass into Highly Porous Ceramics: From Foams to Membranes”, the authors propose a method for recycling medical borosilicate glass (BSG) into adsorbing material capable of wastewater removal. Proposed procedure includes milling of BSG, further activation by strongly alkaline solution leading to partial silica gel-formation and, finally, preparation of membranes by adding surfactant to BSG powder with further annealing. Interestingly, the authors found that the annealing stage can be omitted and the membranes can be obtained even at nearly room temperature (cold consolidation). Finally, they apply these membranes and their composites with crystalline TiO2 for photodegradation of methylene blue.

This paper presents an accurate and skillful study based on an interesting idea of reusing medical glass waste. However, the following issues need to be clarified before the acceptance of the paper.

R3. Obs. #1 Achilles' heel of this study is the BSG source choice. The authors supposed that the proposed procedure could be used to utilize medical glass waste. Actually, such glass waste is heavily polluted by organic biological materials and body liquids. Treating such pollutions with harsh alkaline solutions and annealing them can lead to unforeseen consequences, e.g. burning of organic phase, ash formation, glass cracking, foaming, etc. Such effects can not be considered when using new, fresh and shine medical glass. In turn, recycling of fresh medical-quality glass is nonsense.

A We appreciate the reviewer concern, but there was a misunderstanding. In the present work, ‘fresh’ glass vials were obtained from Stevanato Group. They corresponded to defective vials, not sent to pharmaceutical industries, and abundantly available, considering the huge overall production of vials. The idea behind this work is to set a direction of recycling the specific glass beyond remelting.  

We are actually studying the effect of contaminants in these days. We are discussing with the company the reuse of components made with same glass, but coupled with metals and plastics; the internal crushing facility is currently separating metals and plastic, but the glass residues still has heterogeneities. Current cold consolidation tests do not show any particular effect of these heterogeneities, but they must be properly finalized.

Tests with vials collected from final customers (e.g. hospitals) would be highly interesting. The reviewer is right, organic biological materials and body liquids would be present and represent a scientific challenge. Handling such a waste is actually also a practical challenge, since it is ruled by strict laws. We cannot exclude, however, some work in the future; we are thinking at our process after a low temperature pyrolytic pre-treatment.

R3. Obs. #2 To demonstrate viability of the proposed procedure the authors need to repeat their material preparation process using medical borosilicate glass polluted by some kind of model organic waste or, if possible, using actual waste medical glass.

A We thank the reviewer for the great idea, it could support some research before receiving vials collected from final customers.

 R3. Obs. #3 The authors report full methylene blue (MB) photodegradation by prepared BSG membrane even at the absence of TiO2 Is it MB’s self-photodegradation or membrane itself acts as a photocatalyst or all MB is just adsorbed by the membrane? Please explain.

A We agree with reviewer’s query, Yes MB is fully adsorbed by prepared BSG within 80 minutes. But with the inclusion of TiO2, complete photodegradation was observed within 50 minutes. The composite achieved excellent surface area of 95.5m2/g which boosts the efficiency of the membrane under UV-irradiation, and we achieved better efficiency in less time.

Also, No MB is not self-photodegradation, as we checked the MB sensitivity towards UV degradation, and we don’t observe much change in the absorbance spectrum as shown in Fig. 6a in the manuscript.

R3. Obs. #4 Part “3.1 From foams to membranes” lacks a description of its purpose.

A The part 3.1 is dedicated to elaborating the comparative studies between foamed and unfoamed alkali activated materials, in regards with role of surfactant, temperature and its effects on porosity and mechanical properties. In result, unfoamed samples, exhibiting abundant open porosity, appear as a highly promising material for membranes and finally chosen to be tested to for adsorption of dyes.  Based on these observations, from foams to membranes completely complement with the description.

Round 2

Reviewer 2 Report

My questions were addressed. I recommend its publication without any revision.

Reviewer 3 Report

The authors have addressed all of my comments. The paper is now suitable for publication in Materials.